# In Vitro Study of the Biological and Physical Properties of Dual-Cure Resin-Modified Calcium Silicate-Based Cement

**DOI:** 10.3390/dj11050120

**Published:** 2023-05-04

**Authors:** Minjung Kim, Sung-Hoon Lee, Dong-Hoon Shin

**Affiliations:** 1Department of Conservative Dentistry, College of Dentistry, Dankook University, Cheonan 31116, Republic of Korea; 12180285@dankook.ac.kr; 2Department of Oral Microbiology and Immunology, College of Dentistry, Dankook University, Cheonan 31116, Republic of Korea; dennisyi@dankook.ac.kr

**Keywords:** Theracal PT, Theracal LC, Biodentine, cell viability, odontogenic differentiation, antibacterial activity, microhardness, shear bond strength, pulpotomy

## Abstract

Background: The aim of the present study was to compare the biological and mechanical properties of a novel dual-cure, resin-modified calcium silicate material, Theracal PT^®^ (TP), with those of Theracal LC^®^ (TL) and Biodentine^TM^ (BD). Methods: The cell counting kit-8 was used on human dental pulp cells to test cell the viability of the three materials. Antibacterial activity of TP, TL, and BD against *Enterococcus faecalis* was investigated under anaerobic conditions. The ability of the materials to support odontogenic differentiation was studied by examining the relative gene expression of osteocalcin (OCN), osteopontin (OPN), and Collagen I (ColI) using real-time polymerase chain reaction. For mechanical property tests, microhardness was evaluated using the Vickers microhardness (VHN) test, and the bond strength to the resin was evaluated using a shear bond test machine. Results: There was no significant difference in cell viability between TL and TP after 48 h, and BD showed the highest cell viability, while TP showed the highest antibacterial effect. At the 12-h time point, there was no significant difference in ColI and OCN expression between BD and TP, but TP showed a higher expression of OPN than BD. However, at the 48-h time point, ColI and OCN showed higher levels of expression for BD than for TP and TL. At the same time point, only OPN had a higher diffusion for TP than for BD. TP demonstrated a VHN of approximately 30–35. This value was higher than that of TL and lower than that of BD. In contrast to VHN, the shear bond strength to resin was significantly higher for TL and TP than for BD. Conclusion: TP showed lower biocompatibility than BD but higher OPN expression and antibacterial effects than BD and TL. TP showed higher shear bond strength than BD and higher VHN than TL and BD at the 24-h time point.

## 1. Introduction

For vital pulp therapy, such as pulp capping and pulpotomy, to be successful, adequate materials must be used to form a protective layer over the exposed pulp and maintain pulpal vitality [1,2]. The traditionally suggested and “gold standard” material is calcium hydroxide (Ca(OH)_2_), such as Dycal^®^ (Dentsply Caulk, Milford, DE, USA), which was used a few decades ago [3]. However, the use of Ca(OH)_2_ materials have certain drawbacks, such as poor bonding to dentin, high solubility, the formation of pore-rich dentin bridges, and mechanical instability [4].

Biomaterials such as calcium silicate-based cement and mineral trioxide aggregate (MTA), have been suggested as substitutes for Ca(OH)_2_ materials. MTA has numerous advantages, such as biocompatibility, low solubility, the prevention of bacterial leakage, and the ability to release Ca(OH)_2_ molecules compared with Ca(OH)_2_ materials [5]. However, MTA has some drawbacks, such as its long setting time, difficult handling, and weak physical strength [6,7]. 

To compensate for the disadvantages of MTA, various upgraded materials have been developed. The new upgraded product, Biodentine^TM^ (BD, Septodont, St. Maur-des-Fossés, France), sets with hydration reaction and seems to have improved consistency, handling, and setting time [8]. The manufacturer recommends the placement of composite resin over BD immediately after the initial setting time [9]. However, studies have reported variability in its setting time, which may be longer than that claimed by the manufacturer (12 min) [10]. 

A light-cured, resin-modified calcium silicate material, such as Theracal LC^®^ (TL, Bisco, Inc., Schamburg, IL, USA) is another upgraded biomaterial. It sets with both polymerization and hydration reaction, which makes it easier for handling. It also has the advantage of a higher bond strength to resin composite compared with BD [11]. However, its biological properties are disputed, which hinders its use as a direct pulp capping material [1,12,13]. 

Recently, a novel dual-cure, resin-modified calcium silicate material, Theracal PT^®^ (TP, Bisco, Inc., Schamburg, IL, USA), has been launched in the market for vital pulp therapy. The manufacturer recommends the use of TP for pulpotomy and has mentioned that it is easier to apply and handle than MTA. They have also asserted that it maintains good pulpal vitality by acting as a protectant of the dental pulpal complex [1]. The material is a hydrophilic material and sets with dual polymerization reaction. However, owing to its recent presentation, only a few studies have been conducted on this material [1,14]. 

The aim of the present study was to evaluate the biological and mechanical properties of TP compared with those of TL and BD. The null hypotheses are as follows: (1) there is no difference among the tested materials in terms of cell viability, antibacterial effect, and the odontogenic differentiation potential of human dental pulp cells (hDPCs), and (2) there is no difference between the tested materials in terms of Vickers microhardness (VHN) and bond strength to the composite resin.

## 2. Materials and Methods

### 2.1. Biological Properties

#### 2.1.1. HDPC Culture Preparation

HDPCs (Lonza, Basel, Switzerland) were cultured for cell viability and odontogenic differentiation experiments. Dulbecco’s modified Eagle’s medium (Hyclone, Logan, UT, USA), augmented with 10% fetal bovine serum (FBS), 100 U/mL penicillin, and 100 μg/mL streptomycin sulfate (Gibco Laboratories, Grand Island, NY, USA) was used to culture the hDPCs. HDPCs were cultured in a 100π dish (SPL Lifesciences, Pocheon, Republic of Korea) at 37 °C in a humidified atmosphere containing 5% CO_2_. The cultured cells were transferred to a 24-well plate to perform the biological assays. Passage 10 to 12 of the hDPCs were used in the study.

#### 2.1.2. Cell Viability Assay

Cell viability was assessed using cell counting kit-8 (CCK-8^®^) (Dojindo, Kumamoto, Japan). HDPCs were cultured in a 24-well plate (SPL Life Sciences) at 5 × 10^3^ cells per well with 1 μL medium per well. The plates were incubated until the wells were filled to 70–80%. Next, the test materials were injected into Transwell^®^ inserts (Corning Inc., Lowell, New York, NY, USA) with a bottom microporous membrane (pore size 3.0 μm). The first group was a control group, in which an empty Transwell^®^ insert was placed into the wells (*n* = 8). The second group was the BD group, containing 0.12 g of mixed BD. After insertion, the material was stomped lightly to cover the entire membrane area. BD was mixed according to the manufacturer’s instructions. After insertion, the material was left for 12 min for initial hardening before being placed into the wells (*n* = 8). The third group was the TL group in which 0.12 g of TL was injected into the Transwell membrane (*n* = 8). We ensured that the material covered the entire membrane. TL was light cured (1000 mW/cm^2^, BeLite) for 20 s before being placed into wells. The fourth group was the TP group in which 0.12 g of the material was injected into the Transwell insert. It was then tapped lightly to cover the entire membrane (*n* = 8). The material was light cured (1000 mW/cm^2^, BeLite, B&L, Republic of Korea) for 20 s before being placed in the wells. The plates with the materials were cultured for 24 h and 48 h at 37 °C and 5% CO_2_ under a humidified atmosphere. After 24 h and 48 h of incubation, the transwells were removed. Table 1 shows a list of experimental materials and their compositions. 

The CCK-8 solution was diluted in 1:100. Next, 10 μL of the solution was added to each well and incubated for 1 h. Subsequently, in a 96-well plate, the absorbance was measured at a wavelength of 450 nm, which is indirectly proportional to the number of living cells in the well. Cell viability was expressed as the percentage of absorption for each material, which was compared with the control group. 

#### 2.1.3. Odontogenic Differentiation of the hDPCs

The hDPCs were cultured in 24-well plates, as previously described. The cells were cultured until the wells were filled to 70–80%. The test materials, BD, TL, and TP (0.12 g each), were injected into each Transwell^®^ insert and placed in each well after curing and the initial setting. For the control group, an empty Transwell^®^ was inserted into the wells. The plates were incubated for 12 h and 48 h in the medium at 37 °C, 5% CO_2_, and under a humidified atmosphere. 

After 12 h of incubation in the 12-h groups and 48 h of incubation in the 48-h groups, the materials and medium were removed. To isolate the total RNA from the hDPCs, the protocol recommended by the manufacturer of TRIzol^®^ (Invitrogen Life Tech, Carsbad, CA, USA) was used. TRIzol^®^ (1 mL) was added to each well to release cells from the plate. After incubating for 1–2 min, the reagent containing the cells was transferred to a 1.5 mL microcentrifuge tube (Thermo Fisher, Waltham, MA, USA) and incubated for 5 min. Subsequently, 200 μL of chloroform (Daejung, Siheung, Republic of Korea) was added to the tube. The mixture was vortexed using a vortex mixer for 10 s. The tubes were centrifuged at 12,000× *g* at 4 °C for 15 min. The transparent supernatant (400 μL) was transferred into a new 1.5 mL tube. Isopropyl alcohol (800 μL) was added to the tube and incubated for 1 min. The solutions were centrifuged at 12,000× *g* at 4 °C for 15 min. The supernatant was discarded, and RNA pellets were left at the bottom of the tube. The RNA pellets in each tube were then washed with 1 mL of 75% ethanol. The solution was centrifuged again at 12,000× *g* and 4 °C for 15 min. The supernatant was removed and air-dried. The RNA pellets were then resuspended in 30 μL of 0.1% diethyl pyrocarbonate water (Sigma Aldrich, MI, USA). A nanodrop spectrophotometer (Thermo Fisher Scientific, Wilmington, DE, USA) was used to measure the total RNA concentration in the treated pulp cells.

Complementary DNA was synthesized by mixing 1 μg of total RNA from the hDPCs with the MaximeTM RT-premix kit (iNtRon, Seongnam, Republic of Korea) and incubated at 45 °C for 1 h. To inactivate the reverse transcriptase, the samples were incubated at 95 °C for 5 min. 

Semi-quantitative PCR on an ABI 7500 Real-Time PCR system (Applied Biosystems, Thermo Fisher Scientific) was used to analyze cDNA (2 μL) from hDPCs mixed with 25 μL of 2X SYBR Premix Ex Taq (TB Green^®^, Premix Ex Taq, Takara, Tokyo, Japan), 2 μL of each specific primer (10 μM), and 0.4 μL of 50X ROX reference dye II. The solution was diluted with distilled water to a final volume of 50 μL. Primers for type I collagen (ColI), osteocalcin (OCN), osteopontin (OPN), and glyceraldehyde 3-phosphate dehydrogenase (GAPDH) were used. GAPDH was used as a housekeeping gene to normalize the expression level of the target gene. Table 2 lists the primers used along with their sequences.

Thermal cycling of the final mixture was carried out as follows: an initial denaturation step for 4 min at 95 °C, and thermal cycling steps for 15 s at 95 °C, 15 s at 60 °C, and 33 s at 72 °C. PCR products were analyzed using the amplification of the dissociation curve to investigate the specific amplification of the target gene.

#### 2.1.4. Antibacterial Effect 

An antibacterial activity test was performed according to the method given by the Clinical Laboratory Standard Institute [15]. 

*Enterococcus faecalis* (*E. faecalis*) ATCC 29212 was first cultivated using brain heart infusion broth (BD Biosciences, Franklin Lakes, NJ, USA) under anaerobic conditions (37 °C, 5% H_2_, 10% CO_2_, and 85% N_2_). The concentration of the cultured bacteria was adjusted to 1.0 × 10^6^ cells/mL using a bacterial counting chamber (Marienfeld, Lauda-Konigshöfen, Germany). 

Extracts of BD, TL, and TP were used. Each material was mixed with saline in 15 mL conical tubes at a concentration of 200 mg/mL. Next, they were mixed using a vortex mixer. The mixture was then centrifuged to obtain the extract. The extract was obtained only from the soluble fraction.

In a well on the first row of a 96-well plate, 180 μL of bacteria support media and 180 μL of biomaterial extracts were added. The extracts were two-fold serially diluted using a multi-channel micropipette (Eppendorf, Hamburg, Germany). Next, 20 μL of *E. faecalis* was added to each well and incubated under anaerobic conditions (5% H_2_, 10% CO_2_, and 85% N_2_) at 37 °C for 24 h. A spectrophotometer (Biotek, Winooski, VT, USA) set at 660 nm was used to measure the bacterial growth. The procedure was replicated four times. 

### 2.2. Physical Properties 

#### 2.2.1. Shear Bond Strength Test

Thirty acrylic blocks (2.5 cm diameter × 1 cm height) with a central hole (4 mm diameter × 2 mm depth) were prepared using self-cured acrylic resin (Ortho Jet TM, Lang Dental, Wheeling, IL, USA). Ten blocks were each filled with BD, TL, or TP. BD was prepared according to the manufacturer’s instructions. The mixed material was placed in the acrylic blocks using a metal spatula and pressed flat. The samples were covered with wet cotton pellets and stored in an incubator for 12 min (37 °C, 100% humidity). TL and TP were injected into the holes of the acrylic blocks and light cured with an LED light curing unit (1000 mW/cm^2^, BeLite) for 20 s.

An eighth generation bonding system, All-Bond Universal (Bisco), was applied to the surface of the hardened material. The mixture was agitated for 10 s. After agitation, the surface was air-dried for 10 s and light cured for 20 s. An acrylic mold (Ultradent, South Jordan, UT, USA) was assembled on a bonding clamp with a bonding mold insert to ensure the stability of the sample when applying resin composite to the biomaterials. A flowable resin (Filtek Z350 XT^®^ flow, 3M/ESPE, St. Paul, MN, USA) was injected into the mold to fabricate a 2.38 mm diameter and 2-mm high resin column. The resin was then light cured for 20 s. After polymerization, the samples were removed from the mold and stored for 24 h at 37 °C and 100% humidity. 

After 24 h, the specimens were mounted on a shear bond testing machine (Shear Bond Tester, Bisco). They were subjected to an increasing shear force with a crosshead speed of 0.5 mm/min until bonding failure occurred. 

#### 2.2.2. Vickers Microhardness Test

An acrylic mold with a hole 3 mm in diameter and 1 mm in depth was used. A total of 27 acrylic molds were used. The experimental materials were mixed according to the manufacturer’s instructions. Nine molds were filled with the experimental materials. The specimens were divided into three subgroups: immediately after setting, 24 h after setting, and 7 days after setting. To ensure that the surface of the material was flat, the samples were covered with hard laminate before the initial setting or light curing. TL and TP were light cured using an LED light-curing unit (1000 mW/cm^2^, BeLite) for 20 s. The BD was left to set for 12 min. For the 24-h and 7-day groups, the samples were covered with wet cotton pellets and stored in an incubator (37 °C, 100% humidity). 

VHN was evaluated using a VHN machine (HM-211, Mitutoyo, Kanagawa, Japan). A diamond indenter point weighing 50 g was used for 10 s for the measurements [16]. The Vickers pyramid number (HV) was calculated using the following formula: HV = 1.854 × (F/d^2^), where F is the load in kg^−1^ and d is the mean of the two diagonals produced by the indenter in millimeters [17]. 

### 2.3. Statistical Analysis 

A Shapiro–Wilk test was used to evaluate a normal distribution. For those that had a normal distribution, one-way analysis of variance (ANOVA) was used along with a post hoc Tukey test. For those without a normal distribution, the Kruskal–Wallis and Mann–Whitney tests were used. The significant results were at a 5% level. SPSS 26.0 (SPSS, Chicago, IL, USA) for Windows was used for analysis.

## 3. Results

### 3.1. Biological Properties

#### 3.1.1. Cell Viability Assay

CCK-8 revealed significantly lower cell viability values in the BD group than the control, TL, and TP groups at the 24-h time point (*p* < 0.05) (Figure 1). However, after 48 h, the TL and TP groups showed lower biocompatibility than the control and BD groups (*p* < 0.05). There were no significant differences between the TL and TP groups (Figure 1).

#### 3.1.2. Odontogenic Differentiation

The effects of odontogenic differentiation in the control, BD, TL, and TP groups were evaluated by measuring the expression of odontogenic genes, such as ColI, OCN, and OPN. At the 12-h time point, the control group had a significantly lower expression of ColI, OCN, and OPN than the BD, TL, and TP groups (*p* < 0.05). ColI expression was significantly higher in the TL group than in the TP and BD groups (*p* < 0.05); however, there was no significant difference between the BD and TP groups (Figure 2). No significant difference in OCN expression was found between the test materials at the same time point (Figure 3). OPN expression was highest in the TL and TP groups, followed by the BD and control groups (*p* < 0.05). No significant differences were found between the TL and TP groups (Figure 4).

At the 48-h time point, significant differences were found among all groups, wherein the BD group had the highest ColI expression, followed by the TL, control, and TP groups (*p* < 0.05) (Figure 2). Regarding OCN expression, BD and TL expressed significantly more OCN than the control group (*p* < 0.05). However, TP showed no significant difference in OCN expression compared with the control group (Figure 3). There was a significant difference in the expression level of OPN between all groups, wherein TP had the highest OPN expression and the control group had the lowest OPN expression (*p* < 0.05) (Figure 4).

#### 3.1.3. Antibacterial Effect 

A direct correlation was found between all test materials and their concentrations (*p* < 0.05) (Figure 5). A significant difference was found at 25 mg/mL, 50 mg/mL, and 100 mg/mL of the diluted materials, where TP showed the highest antibacterial effect, and the control group showed the lowest antibacterial effect (*p* < 0.05). The TP, TL, BD, and control groups followed in sequence.

### 3.2. Physical Properties 

#### 3.2.1. Shear Bond Strength

The materials tested using the Shapiro–Wilk test showed no normality at *p* < 0.05. The BD group had the lowest shear bond strength (*p* < 0.05); however, no significant difference was found between the TL and TP groups (Figure 6).

#### 3.2.2. Vickers Microhardness

Immediately after setting, there was a significant difference in VHN between the material groups, measured using the Kruskal–Wallis test, where the TP group had the highest VHN and the BD group had the lowest VHN (*p* < 0.05). After 24 h, no significant differences were found between the groups. After 1 week, in contrast to the first and second time points, the BD group had the highest VHN, while the TL group had the lowest VHN (*p* < 0.05). The TL and TP groups showed no significant differences in terms of the setting time. In contrast, BD showed a significant difference with regard to the setting time and showed the highest microhardness 1 week after the setting (*p* < 0.05) (Figure 7).

## 4. Discussion

This study investigated the biological and physical properties of three calcium silicate cements, namely, BD, TL, and TP, which are important for evaluating the suitability of these materials for vital pulpal therapy. 

A biocompatibility evaluation is important because the material is in direct contact with pulp tissue during vital pulp treatment. Non-biocompatibility results in an eventual loss of the vitality of pulp tissue [18]. HDPCs are one of the first cells that come in direct contact with the material; therefore, the use of these cells is crucial in biocompatibility tests [19,20]. 

The benefit of this test is its controlled design and evaluation [21]. Biocompatibility assays can be divided into two types: direct and indirect cell–material contact. The method chosen for the present study is the Millipore filter technique, which is an indirect cell–material contact method [22]. This method induces better cell growth than the direct contact method [23]. Freshly mixed and set materials yield different biocompatibility results. However, we used freshly set material because it is comparable to a clinical situation.

The present study revealed a higher biocompatibility for BD than for TL and TP. Although TP had a higher biocompatibility than TL, there was no significant difference at the 48-h time point. Our verification agrees with a previous study that compared MTA Angelus^®^, TL, and TP [1]. Interestingly, BD showed the lowest cell viability at the 24-h time point; however, it increased at the 48-h time point. In contrast, the cell viability of TL and TP reduced from 24 to 48 h. Poggio et al. revealed the same alteration in cell viability for TL as in the present study [4]; however, Adiguzel et al. demonstrated a contrary result, wherein cell viability increased with time [24]. Different results may be attributed to differences in sample preparation methods, as Adiguzel et al. used initially cured disc materials for the test, whereas Poggio et al. used the Transwell method as in the present study [4,24].

The inclusion of resin monomers in TL and TP could be the reason for the lower cell viability. Unpolymerized monomers may remain after contact with pulp tissue [12,13], even after polymerization. Although the actual degree of conversion of TP compared to TL is unknown, dual-cure materials have a higher degree of conversion than light-cured materials [25].

For successful vital pulp therapy, the bioactive materials used should also help the pulpal cells to proliferate and differentiate into odontoblast-like cells. Studies have shown that hDPCs can be used to investigate odontogenic differentiation [26]. Other studies have shown that cultured hDPCs can be induced to differentiate into odontoblast-like cells in culture dishes [27,28]. This can be assessed through the expression of different gene markers such as OPN (gene name SPP1), ColI (gene name COL1A1), and OCN (gene name BGLAP), which were used to explore the differentiation and mineralization of osteoblasts and odontoblasts in the present study [29,30]. 

An interesting finding in the present study is that BD had high OCN and ColI expression values at 12 h, and similar results were found at 48 h. Furthermore, the gene expression of ColI and OCN in TL and TP downregulated from 12 h to 48 h. This could indicate that the odontogenic differentiation for TL and TP was less than that for BD. Our ColI expression result agrees with a previous study by Rodriguez-Lozano et al., where MTA Angelus showed the highest ColI expression at 7 days [1] but disagrees with a study by Sanz et al., where BD and TP showed lower expressions of ColI than the control [30]. Sanz et al. also showed a higher ColI expression for TP than BD [30]. ColI is a marker related to the early stage of odontogenic differentiation [31]; moreover, it accelerates odontogenic differentiation and mineralization of stem cells [32]. In contrast, OCN is a marker related to the late stage of odontogenic differentiation and is also a well-characterized bone protein [30,33]. 

In contrast to ColI and OCN, TP showed higher OPN gene expression than BD at 12 h. The expression upregulated in TL and TP at 48 h compared to BD. This finding contrasts with a study by Kwon et al., in which resin monomers caused the downregulation of OPN and other odontogenic/osteogenic gene markers [34]. OPN is a gene marker that is a phosphorylated glycoprotein, which is essential for type I collagen secretion by newly differentiated odontoblast-like cells [35]. Therefore, it is found in the extracellular matrix of bones and teeth and is related to hard-tissue mineralization [36]. OPN is also expressed by fibroblasts and is associated with wound healing [37]. Overall, the expression levels of OPN were higher than those of ColI and OCN. High OPN expression levels compared to other genes can be explained by the ability of cultured hDPCs to express the OPN gene in high levels [38] or during the short experimental period [31]. 

For the pulpal tissue to heal in the right manner, the underlying tissue should be well protected from oral microorganisms. This can be achieved through proper sealing and antibacterial activity [39,40]. Therefore, the antibacterial activities of the materials were analyzed. TP had the highest antibacterial activity, whereas BD had the lowest. The antibacterial activity of calcium silicate cement has been hypothesized to be due to its high pH ranging between 11 and 12 [41]. The pH of BD seemed to be higher than that of MTA [42] and TL, with a pH ranging between 8 and 10 [43,44]. According to Elbanna et al., the pH of TP was lower than that of TL (TL = 8.9, TP = 7.8) [45]. Based on this information and our present study, pH may not be related to antibacterial effects.

*E. faecalis* is an oral microorganism that forms compact biofilms and has a good ability to survive in unfavorable conditions [46]. *E. faecalis* has difficulty surviving at a pH ≥ 11.5 [47]. Our present study demonstrated the high antibacterial activity of TP and the low antibacterial effect of BD. This indicates that there are factors other than high pH that may be related to the antibacterial effect. According to a previous study by Rodriguez-Lozano et al., TP and TL released more Ca^2+^ than MTA, which may have influenced the antimicrobial effect [1]. In contrast to other studies [4,48], the present study used uncured/unset material diluted in saline. This may have caused the unpolymerized monomers to remain in the resin-modified calcium silicate cement [49]. 

For a dental base material to resist occlusal stress and deformation, its hardness should be adequate [50]. The best state of microhardness is similar to that of dentin itself. The ability of dentin to partially disperse and withstand occlusal force can prevent local deformation, and this can be measured using VHN. For the base materials used in vital pulp therapy to be tightly sealed and withstand the overlying restoration and occlusal force, they should have a microhardness similar to that of dentin [51,52]. The mean VHN value for sound human dentin is reported to be in the range of 57–62 [52,53]. The present study revealed that none of the materials had VHN values greater than 36 until 24 h after placement. However, the mean VHN for BD after 1 week was 56.8, which is similar to the VHN of sound dentin. A previous study by Kaup et al. reported that the VHN of BD was 62 after 2 h [51]. This could result from the difference in experimental designs, that is, the thickness of the material used. Unlike BD, the mean VHN of TP was the same from immediately after setting to 1 week after setting and remained approximately 30. However, TL had a much lower VHN, approximately between 10 and 15, even after 1 week. Although TL is insufficient for use as a restorative base, TP has a high VHN at an early time point, and it may have a positive meaning as a base. Although only VHN was observed in the present study, material mechanical properties should be interpreted after performing further physical tests. 

As bioceramic materials can be successfully used in vital pulp therapy, it is important for them to be tightly bonded to overlying restorations, such as composite resins [54]. A minimum of 17–20 MPa bond strength is required to withstand shrinkage and the formation of marginal gaps [55,56]. As stated by the manufacturer of BD, the BD product should be initially set at approximately 10–12 min, and the composite resin should be placed above it after 12 min. TL contains resin monomers and calcium silicate powder [44]. In a previous study by Karadas et al., TL was found to bind well to composite resin [57]. 

To imitate a busy clinical situation and the use of bioceramic materials, universal bonding was applied immediately after setting (12 min after setting for BD and after light curing for TL and TP). The present study revealed a high shear bond strength for both TL and TP. However, BD showed an unsatisfactory bond strength with the composite resin. Some of the composite resins became separated from the BD specimens immediately after the setting. The elevated bond strength for TL and TP may be ascribed to the existence of dimethacrylate monomers that advance chemical adhesion between the resin adhesive and resin-modified calcium silicate cements [57]. The bonding mechanism between BD and the resin adhesive is yet to be determined. Nevertheless, previous studies have reported a lower bond strength between BD and resin composite after 12 min than after 24 h or longer. These studies demonstrated that, the longer BD takes to mature, the better the bond strength [58,59]. 

Based on the present study on physical properties, it can be concluded that BD should not be restored directly on the same day the material is used. However, after at least one day, it can be restored with the resin composite on the material itself. In contrast, resin composite restoration could be used over TP and TL on the same day as vital pulp therapy. 

From the results of this study, the null hypothesis that there is no difference between BD, TL, and TP in cell viability, antibacterial effect, odontogenic differentiation potential on hDPCs, VHN, and bond strength to composite resin was rejected.

## 5. Conclusions

TP showed lower cell viability than BD and lower ColI and OCN expression than BD and TL, which could be a negative property of TP. However, it showed higher OPN expression and higher antibacterial effects than TL and BD, which could be beneficial biological features. TP showed a higher shear bond strength than BD, and a higher VHN than TL and BD at 24 h. 

From the above results, it can be concluded that TP may be potentially used in vital pulpal tissue treatments. 

## Figures and Tables

**Figure 1 dentistry-11-00120-f001:**
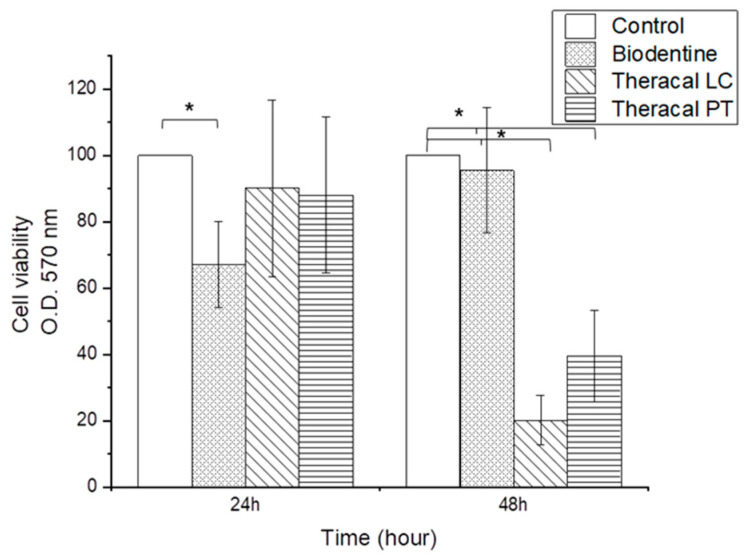
Cell viability of hDPCs treated with BD, TL, and TP. Asterisk indicating significant difference between the experimental groups (*p* < 0.05).

**Figure 2 dentistry-11-00120-f002:**
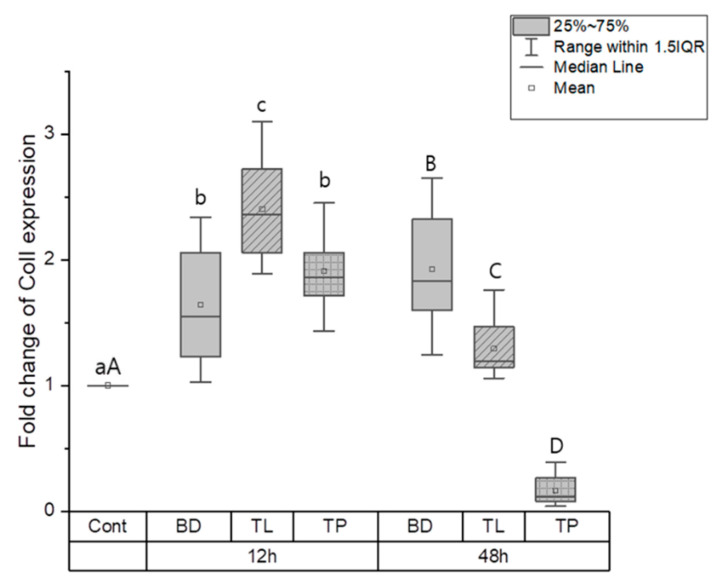
Collagen I gene expression for BD, TL, and TP. Different lower case letters indicate a significant difference between groups at the 12-h time point (*p* < 0.05). Different upper case letters indicate a significant difference between groups at the 48-h time point (*p* < 0.05). At the 12-h time point, the TL group showed significantly higher Col1 expression compared to the control, BD, and TP groups (*p* < 0.05). At the 48-h time point, the BD group showed the highest and TP group the lowest ColI expression (*p* < 0.05).

**Figure 3 dentistry-11-00120-f003:**
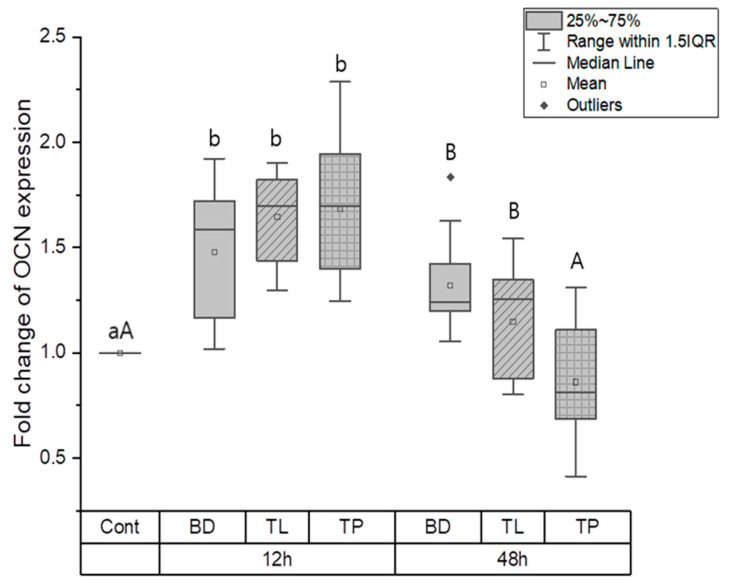
Osteocalcin gene expression for BD, TL, and TP. Different lower case letters indicate a significant difference between the groups at the 12-h time point (*p* < 0.05). Different upper case letters indicate a significant difference between the groups at the 48-h time point (*p* < 0.05). At the 12-h time point, the BD, TL, and TP groups showed significantly higher OCN expression compared to the control group (*p* < 0.05). At the 48-h time point, BD and TL showed significantly higher OCN expression compared to the control and TP groups (*p* < 0.05).

**Figure 4 dentistry-11-00120-f004:**
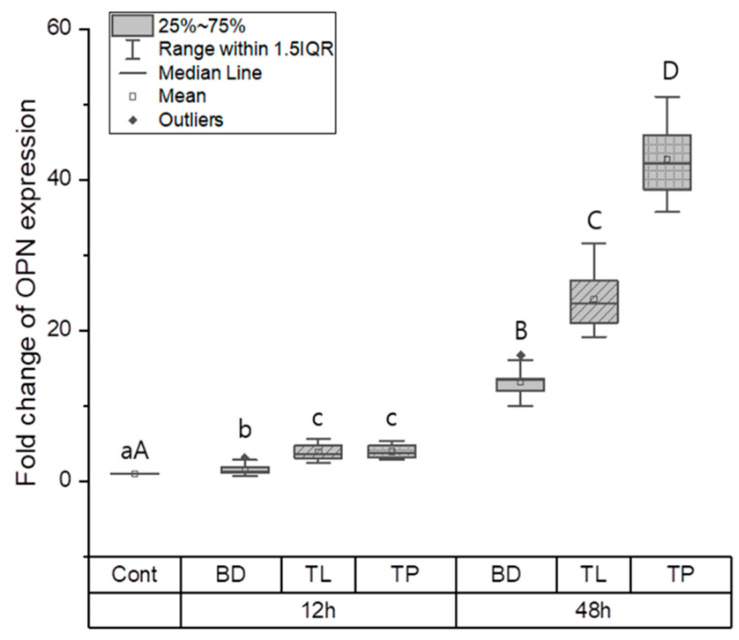
Osteopontin gene expression for BD, TL, and TP. Different lower case letters indicate a significant difference between the groups at the 12-h time point (*p* < 0.05). Different upper case letters indicate a significant difference between the groups at the 48-h time point (*p* < 0.05). At the 12-h time point, TP and TL showed significantly higher OPN expression compared to the control and BD groups (*p* < 0.05). At the 48-h time point, TP showed the highest OPN expression (*p* < 0.05).

**Figure 5 dentistry-11-00120-f005:**
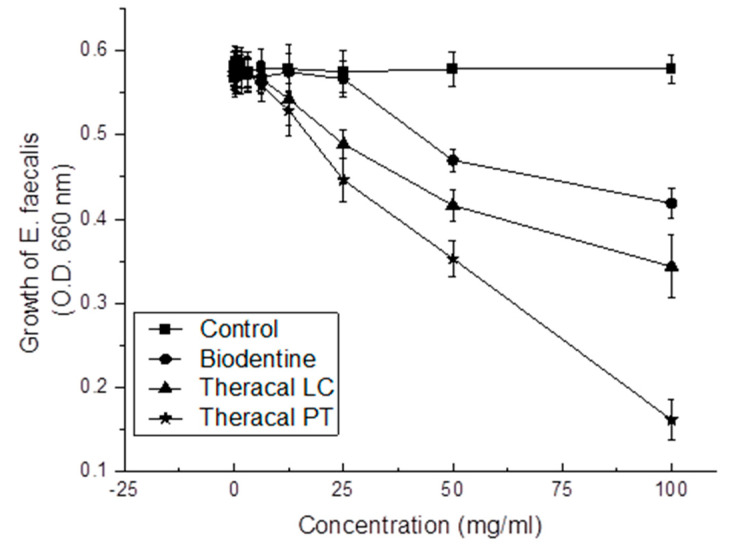
Antibacterial activity of BD, TL, and TP against *E. faecalis*. The TP group showed the highest level of antibacterial activity compared to the control, BD, and TL groups (*p* < 0.05).

**Figure 6 dentistry-11-00120-f006:**
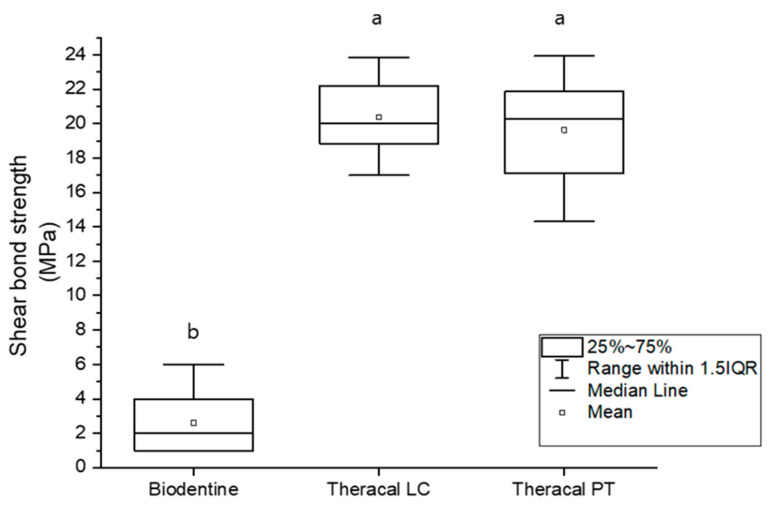
The shear bond strength of BD, TL, and TP to resin composite. Different lower case letters indicate a significant difference between the experimental groups (*p* < 0.05). The graph shows the significantly lower shear bond strength of BD compared to the TL and TP groups (*p* < 0.05).

**Figure 7 dentistry-11-00120-f007:**
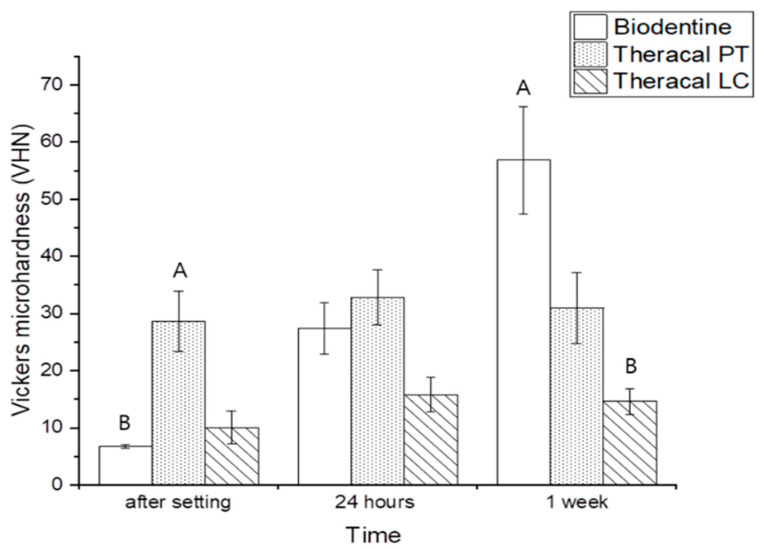
Vickers microhardness of BD, TL, and TP: A: Showing the highest significant group at each time point with a Kruskal–Wallis test. B: Showing the lowest significant group at each time point with a Kruskal–Wallis test. Kruskal–Wallis test showed a significant difference between the experimental groups (*p* < 0.0.5), where TP had the highest and the BD group the lowest VHN immediately after the setting. In contrast, BD had the highest, and the TL group the lowest VHN after 1 week of the setting (*p* < 0.05).

**Table 1 dentistry-11-00120-t001:** List of the materials used in the experiments.

Material Type	Product Name	Compositions	Manufacturer
Tricalcium silicate cement	Biodentine^TM^	Powder: tricalcium silicate, calcium carbonate, and zirconium dioxide Liquid: water, and calcium chloride	Septodont, St. Maur-des-Fossés, France
Light cure resin-modified calcium silicate cement	Theracal LC^®^	Portland cement type III (20–60%), poly(ethylene glycol) dimethacrylate (10–50%), bis-GMA (5–20%), and barium zirconate (1–10%)	Bisco, Inc., Schamburg, IL, USA
Dual cure resin-modified calcium silicate cement	Theracal PT^®^	Base: SG-Mix cement (50–75%), polyethylene glycol dimethacrylate (10–30%), bis-GMA (5–10%), and barium zirconate (1–5%) Catalyst: barium zirconate (1–5%), ytterbium fluoride (1–5%), and initiator (<1%).	Bisco, Inc., Schamburg, IL, USA

**Table 2 dentistry-11-00120-t002:** The primers used for real-time PCR and their primer sequence.

Primer		Sequence
OCN	F	5′-CGG TGC AGA GTC CAG CAA AG-3′
	R	5′-TAC AGG TAG CGC CTG GGT CT-3′
OPN	F	5′-ACA CAT ATG ATG GCC GAG GTG A-3′
	R	5′-GTG AGG TGA TGT CCT CGT CTG TAG-3′
ColI	F	5′-CTG CTG GAC GTC CTG GTG AA-3′
	R	5′-ACG CTG TCC AGC AAT ACC TTG A-3′
GAPDH	F	5′-GTG GTG GAC CTG ACC TGC-3′
	R	5′-TGA GCT TGA CAA AGT GGT CG-3′

OCN: osteocalcin, OPN: osteopontin, ColI: type I collagen, GAPDH: glyceraldehyde 3-phosphate dehydrogenase.

## Data Availability

Data are available upon reasonable request.

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
