# Peer review of "In Vitro Study of the Biological and Physical Properties of Dual-Cure Resin-Modified Calcium Silicate-Based Cement"

_dentistry, 2023, doi:10.3390/dj11050120_

Round 1

Reviewer 1 Report

Dear Authors. Thank You for the well-written article. Just few comments:

Introduction

Calcium hydroxide as the gold standard for pulp capping or pulpotomy was used a few decades ago;  nowadays we use “bioceramic” materials. You should make corrections, letting readers understand it’s the past.

Methods

WHY was the LOT # added to Table 1? It is not necessary and doesn't provide any real and useful information.

Discussion

Check how E. faecalis should be written. E. faecalis

Conclusions

You should NOT repeat that “TP is a dual-cured, resin-modified calcium silicate material”; it was provided in the introduction, and it is NOT a matter of conclusions at all.

I would suggest rephrasing “From the above results, it can be concluded that TP can be used carefully in vital pulpal tissue treatments.” Not carefully, but maybe potentially can be used, safely used article. 

There are several comments corresponding to some questions. Please check it in the attachment.

Author Response

Thank you for your comments. I have made the modifications. 

Introduction 

For vital pulp therapy, such as pulp capping and pulpotomy, to be successful, adequate materials must be used to form a protective layer over the exposed pulp and maintain pulpal vitality [1, 2]. The traditionally suggested and “gold standard” material is calcium hydroxide (Ca(OH)2), such as Dycal® (Dentsply Caulk, Milford, DE, USA), which was used a few decades ago [3]. However, the use of Ca(OH)2 materials have certain drawbacks, such as poor bonding to dentin, high solubility, formation of pore-rich dentin bridges, and mechanical instability [4].  

Biomaterials such as, Calcium silicate-based cement and mineral trioxide aggregate (MTA), have been suggested as substitutes for Ca(OH)2 materials. MTA has numerous advantages, such as biocompatibility, low solubility, prevention of bacterial leakage, and the ability to release Ca(OH)2 compared with Ca(OH)2 materials [5].

Methods 

Lot # are deleted. 

All Enterococcus faecalis and E. faecalis are changed to Enterococcus faecalis or E. faecalis

Conclusions

The sentence TP is a dual-cured, resin-modified calcium silicate material” is deleted. 

“From the above results, it can be concluded that TP can be used carefully in vital pulpal tissue treatments.” is changed to “From the above results, it can be concluded that TP maybe potentially used in vital pulpal tissue treatments.”

Reviewer 2 Report

The manuscript describes the examination of the biological and mechanical properties of a novel dual-cure resin-modified calcium silicate material, and comparison with Theracal PT®(TP) with those of Theracal LC®(TL) and BiodentineTM (BD).

The scope is moderate, the methods are appropriate and the results are sufficient to support the conclusions.

The setting mechanism of all three materials should be introduced.  

Hence, a minor revision is necessary.

The following points are to be addressed:

1. Moderate English checks, L42,

2. L57, some relevant studies of TP should be introduced here.

3. L69, move donation to Acknowledgement.

4. The sample storage conditions are missing for Vickers hardness test.

5. The significance of letters, big and small a,b,c should be stated in each figure caption.

6. Statement “Biocompatibility evaluation is important because the material is in direct contact with pulp tissue during vital pulp treatment. Non-biocompatibility results in eventual loss of vitality of pulp tissue.” Needs a reference. This reviewer suggests doi: 10.1016/j.tiv.2019.104627

7. Author contribution, Acknowledgement sections are missing. There may be more sections please check journal requirement.

Author Response

The setting mechanisms for all three materials are added: 

  • Biodentine: L47
  • Theracal LC: L54
  • Theracal PT: L59
  1. I am not sure about the english check. But,

    " MTA has numerous advantages, such as biocompatibility, low solubility, prevention of bacterial leakage, and the ability to release Ca(OH)2 compared with Ca(OH)2 materials" to 

    "MTA has numerous advantages, such as biocompatibility, low solubility, prevention of bacterial leakage, and the ability to release Ca(OH)2 molecules compared with Ca(OH)2 materials"

  2. The studies are introduced by adding references, line 61.
  3. The sentence about the donation of the cells are moved to 'acknowledgement' 
  4. The sentence about how the samples are stored at Vickers hardness test are added on the manuscript. 
  5. The significance of lower and upper case letters are mentioned on figure legends 
  6. Reference is added [17]
  7. Patents sections are checked and added to the manuscript

The manuscript itself is revised and is attached. 
